

# A synthetic biosensor to detect peroxisomal acetyl-CoA concentration for compartmentalized metabolic engineering

Herbert M. Huttanus[1] and Ryan S. Senger[1,2]

[1] Department of Biological Systems Engineering, Virginia Polytechnic Institute and State University (Virginia Tech), Blacksburg, VA, United States of America
[2] Department of Chemical Engineering, Virginia Polytechnic Institute and State University (Virginia Tech), Blacksburg, VA, United States of America

## ABSTRACT

**Background**. Sub-cellular compartmentalization is used by cells to create favorable microenvironments for various metabolic reactions. These compartments concentrate enzymes, separate competing metabolic reactions, and isolate toxic intermediates. Such advantages have been recently harnessed by metabolic engineers to improve the production of various high-value chemicals via compartmentalized metabolic engineering. However, measuring sub-cellular concentrations of key metabolites represents a grand challenge for compartmentalized metabolic engineering.

**Methods**. To this end, we developed a synthetic biosensor to measure a key metabolite, acetyl-CoA, in a representative compartment of yeast, the peroxisome. This synthetic biosensor uses enzyme re-localization via PTS1 signal peptides to construct a metabolic pathway in the peroxisome which converts acetyl-CoA to polyhydroxybutyrate (PHB) via three enzymes. The PHB is then quantified by HPLC.

**Results**. The biosensor demonstrated the difference in relative peroxisomal acetyl-CoA availability under various culture conditions and was also applied to screening a library of single knockout yeast mutants. The screening identified several mutants with drastically reduced peroxisomal acetyl-CoA and one with potentially increased levels. We expect our synthetic biosensors can be widely used to investigate sub-cellular metabolism and facilitate the "design-build-test" cycle of compartmentalized metabolic engineering.

# INTRODUCTION

Sub-cellular compartmentalization is used by all eukaryotes and some prokaryotes as a means to create favorable micro-environments for various metabolic reactions. These compartments concentrate enzymes and substrates (*Choi & Montemagno, 2006*), increasing the rate of reaction. Compartmentalization also helps to keep incompatible pathways separate (*Zecchin et al., 2015*) and can provide a safe area for toxic intermediates. Recently, metabolic engineers have explored the concept of pathway compartmentalization to

Corresponding author
Ryan S. Senger, senger@vt.edu

enhance the performance of metabolic pathways (*Avalos, Fink & Stephanopoulos, 2013*; *Zecchin et al., 2015*; *Zhou et al., 2016*).

One challenge to the continued progress of compartmentalized metabolic engineering is the lack of tools for measuring local metabolite concentrations at a sub-cellular scale. There are many proprietary colorimetric and fluorescent assays (such as glucose peroxidase assays (*Ngo & Lenhoff, 1980*) or enzymatic acetyl-CoA assays (*Fritz et al., 2013*)) but these *in vitro* methods often require cell lysis. Cell fractionation involves separating cell compartments of interest beforehand, but contamination with similarly sized organelles is possible following differential or density gradient centrifugation (*Kikuchi et al., 2004*). In vivo biosensors are another popular method for screening the response of metabolites to various perturbations. These techniques often involve the use of transcription factors that recognize the analyte and activate bioluminescence or fluorescence reporter genes (*Michener et al., 2012*; *Mustafi et al., 2012*; *Brognaux et al., 2013*; *Li & Yu, 2015*; *Mahr et al., 2016*; *Skjoedt et al., 2016*). This, generally, means that the detection of the analyte largely occurs in the cytoplasm or nucleus.

To provide a tool for measuring metabolite availability in sub-cellular compartments, we propose using a localized synthetic biosensor as a form of metabolic assay. In this method, heterologous enzymes are expressed with localization tags to produce some reporter compound only in the compartment of interest. With sufficiently high levels of enzyme expression, the production of the reporter compound would be rate-limited by the substrate concentration, allowing for relative quantification of the substrate. Enzymes should be selected that use either no substrates besides the metabolite of interest or that use cofactors that are much more available than the metabolite of interest such that the cofactors do not become limiting. The reporter compound can then be extracted from the cell and quantified. Herein, we showcase the use of one such synthetic biosensor to measure acetyl-CoA in the peroxisomes of the yeast *Saccharomyces cerevisiae*. Acetyl-CoA is a core metabolite whose availability has implications in many important metabolic pathways. Measuring total, cellular acetyl-CoA levels is a simple matter using commercially available enzymatic assay kits (*Jose & Suraishkumar, 2016*), but measuring acetyl-CoA in organelles is more challenging since conventional assays would require the organelles to first be isolated by cell fractionation. Polyhydroxybutyrate (PHB) production can also be used as an indirect marker of acetyl-CoA availability. Only three enzymes are needed for the conversion of acetyl-CoA into PHB (*Wang & Yu, 2007*) and they can be efficiently localized to the peroxisome by inclusion of C-terminal PTS1 peptide tags (*Kim & Hettema, 2015*). Acetyl-CoA does not readily cross membranes (*Chen, Siewers & Nielsen, 2012*), so the peroxisome-localized enzymes should only have access to the peroxisomal pool of acetyl-CoA (Fig. 1). In this paper, a localized enzymatic assay and HPLC analysis of the cellular PHB content were used to qualify the response of acetyl-CoA in the peroxisome to several culturing conditions and gene knockout perturbations.

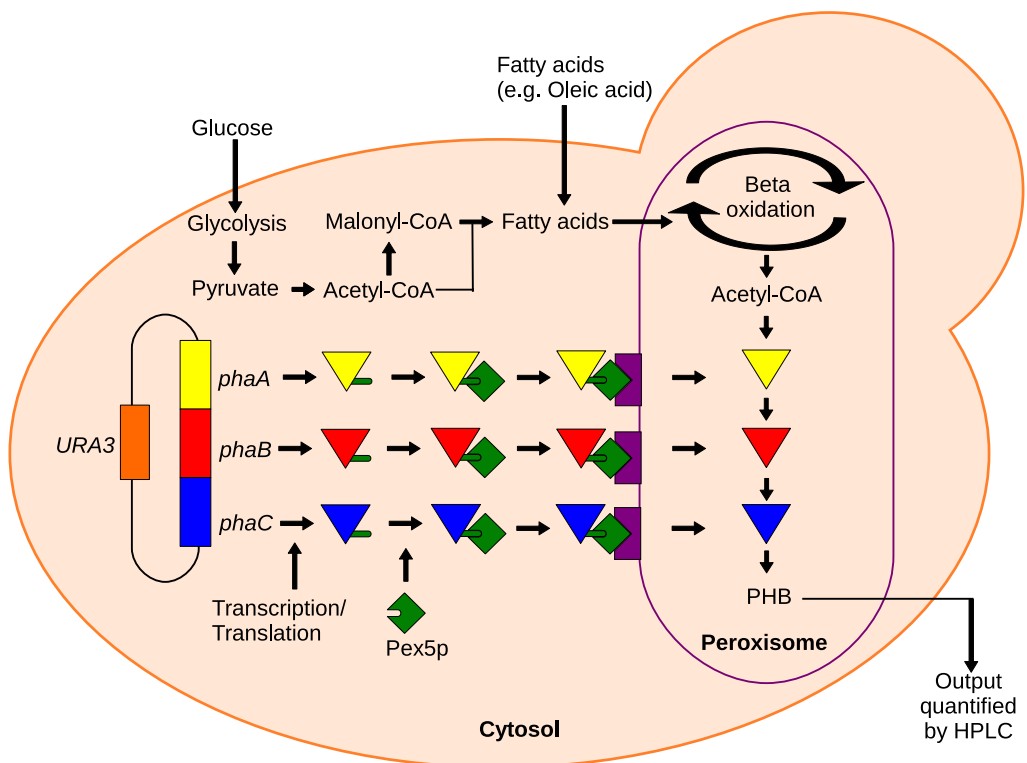

**Figure 1  Mechanism of the localized biosensor.** The three PHB enzymes were expressed from a plasmid and contained enhanced PTS1 localization tags. The tags were recognized by peroxisomal targeting signal receptor Pex5p which shuttles the enzymes into the peroxisome. There, the enzymes converted peroxisomal acetyl-CoA into PHB that was then extracted and detected by HPLC.

## MATERIAL AND METHODS

### Media and strains

Minimal media was prepared as previously reported (*Blank & Sauer, 2004*; *Guo et al., 2016*) with a glucose concentration of 0.5% w/v. Yeast nitrogenous media YN contains 0.67% w/v yeast nitro base, 0.1% yeast extract and 0.5% glucose. Synthetic complete media contained 0.17% yeast nitro base, 0.5% ammonium sulfate, 0.5% glucose and 1x CSM Ura supplement from MP biomedicals. Oleic acid was added to the media in some cases to enhance acetyl-CoA availability in the peroxisome through beta oxidation. All cultures were incubated at 30 °C in a shaking incubator. The experiment testing for the effect of media composition used 10 mL cultures each and the mutant screening cultures were 3 mL cultures each from which only 2 mL was harvested at the end of culturing. The growth/production curve experiments used 20 mL cultures from which 2 mL were drawn at each time point. All of the homologous recombinations were performed in *S. cerevisiae* Invitrogen strain INVSc1. Plasmids were transferred to *E. coli* strain Top10 for verification and long-term storage. All cultures used to collect and measure PHB were performed in *S. cerevisiae* strain BY4741 or BY4741-derived single gene knockouts.

**Table 1  Plasmids and strains used in this study.** Plasmids and strains ending in the letter "c" for "cytosolic" express heterologous PHB enzymes without localization tags. Plasmids and strains ending in "p" for "peroxisomal" express enzymes modified with enhanced Peroxisome Targeting Sequences (ePTS1).

**Plasmid**

| Name | Description | |
|------|-------------|---|
| pPHBc | pRS416-(adh1t-phaA-pgk1p)-(tef1p-phaB-cyc1t)-(pdc1p-phaC-pdc1t) | |
| pPHBp | pRS416-(adh1t-ePTS1-phaA-pgk1p)-(tef1p-phaB-ePTS1-cyc1t)-(pdc1p-phaC-ePTS1 pdc1t) | |

**Strains**

| Name | Phenotype | Plasmid |
|------|-----------|---------|
| BY4741 | MATa his3Δ1 leu2Δ0 met15Δ0 ura3Δ0 | |
| phbC | Same as BY4741 | pPHBc |
| phbP | Same as BY4741 | pPHBp |

## Cloning

The major plasmids and strains are shown in Table 1. Other strains and plasmids are detailed in Supplementary Methods S1. The three *pha* genes were codon-optimized for yeast and kindly provided by Jens Nielsen (*Kocharin et al., 2012*). Construction of the plasmids pPHBc and pPHBp was accomplished by an established, yeast homologous recombination-based method, DNA assembler (*Shao & Zhao, 2009*). Briefly, DNA fragments were amplified by PCR using primers that add homologous overlap between adjacent fragments. The fragments were then co-transformed into *S. cerevisiae* along with a linearized backbone to assemble elements in a single step (*Shao, Luo & Zhao, 2011*). Gene fragments used to construct plasmid pPHBp used modified primers for the addition of ePTS1 localization tags (*DeLoache, Russ & Dueber, 2016*; *Liu & Naismith, 2008*). Additional methods details are provided in Supplementary Methods S1.

## HPLC analysis

Culture samples of 10 mL were centrifuged at 4000 rpm for five minutes, and rinsed twice with 10 mL of $ddH_2O$. Pellets were then dried for 48 hours at 70 °C. PHB is insoluble, so it is regularly converted to crotonic acid monomers by sulfuric acid digestion (*Karr, Waters & Emerich, 1983*). The pulverized dry pellets (or thin films for smaller samples) were digested with 100 μL of concentrated $H_2SO_4$ per mL of the pre-dried sample. Acid digests were carried out at 95 °C for 1 hour. The digest was diluted with 400 μL of $ddH_2O$ for every 100 μL of acid used. The carbonified cell debris was removed by centrifugation at 13,000 rpm for 10 min. These samples were analyzed by HPLC with an Aminex HPX-87H ion exclusion column with a temperature of 60 °C and a flow rate of 0.6 mL min-1 and with 5 mM $H_2SO_4$ as the mobile phase.

# RESULTS

## Dependence on fatty acids

To test the viability of the localized synthetic biosensor, a proof-of-concept study was performed in which yeast cells with PHB-producing genes (with or without peroxisomal localization) were analyzed to compare acetyl-CoA levels in the cytosol and peroxisome.

PHB production was achieved by expressing three enzymes from *Ralstonia eutropha* H16. The first enzyme, acetyl-CoA acetyltransferase [EC 2.3.1.9], consumes two acetyl-CoA to produce acetoacetyl-CoA. The next enzyme, NADPH-dependent acetoacetyl-CoA reductase [EC 1.1.1.36] reduces the acetoacetyl-CoA in order to generate 3-hydroxybutyric-CoA monomers. The monomers were then polymerized by the final enzyme, PHB synthase [EC 2.3.1.-] (*Wang & Yu, 2007*). These three enzymes are encoded by the genes *phaA*, *phaB* and *phaC* respectively. The genes were previously codon-optimized for use in yeast cells (*Kocharin et al., 2012*). All three genes were combined in each of two plasmids. Plasmid pPHBc contained the codon-optimized genes without peroxisomal localization tags and thus the enzymes localize in the cytoplasm. Plasmid pPHBp contains the same three enzymes but with peroxisomal localization tags designed for rapid and efficient import (*DeLoache, Russ & Dueber, 2016*) which were added by homologous recombination (*Joska et al., 2014*). We chose PHB as our reporter compound because of two considerations. First, PHB is detected relatively easily and is non-native to yeast cells. Second, the PHB pathway is relatively short with only three enzymes, which reduces the risk of metabolic burden. One limitation of this biosensor is that the PHB production pathway also uses NADPH as a cofactor allowing for the possibility that NADPH will become the limiting reagent at sufficiently high acetyl-CoA levels.

To show that peroxisomal acetyl-CoA could be used for PHB production, we cultured the peroxisomal and cytosolic variants of the PHB producing strains (phbP and phbC respectively) in various media with various external concentrations of oleic acid for 48 hours. Three media were used: synthetic complete yeast media (SC), minimal media, and yeast nitro base (YN) media. Peroxisomal levels of acetyl-CoA were enhanced by the addition of oleic acid which, as a fatty acid, is broken down through beta oxidation in the yeast peroxisome to yield acetyl-CoA (*Chen, Siewers & Nielsen, 2012*). Oleic acid content ranging from 0%, 0.2% and 0.5% w/v was added into each of the media. The cytosolic *pha* genes were able to produce similar levels of PHB with any of the media and at all of the oleic acid concentrations present (Fig. 2A). No PHB was produced by the peroxisomal strain phbP when oleic acid was absent from the media (Fig. 2B). Also, we found that with the increase of oleic acid concentration in the YN media, the PHB production also increased.

## Growth and production curves

Both strains were then cultured in YN media with 0.5% oleic acid and sampled periodically until PHB production plateaued (Fig. 3). Strain phbC achieved a maximal PHB level of around 30 mg/g DCW within only 24 hours, which was the same amount of time it took to achieve a maximum cell density (~5 O.D.). Strain phbP reached the same cell density in those 24 hours, but only started producing PHB after that point. PHB production in the phbP strain reached a maximum of about 7 mg/g DCW after five days.

## Knockout screening

We demonstrated the application of our compartmentalized biosensor by screening several mutants for altered levels of peroxisomal acetyl-CoA. In brief, thirteen yeast single knockout mutants were selected and transformed with pPHBp. Three of the genes, *adr1*,

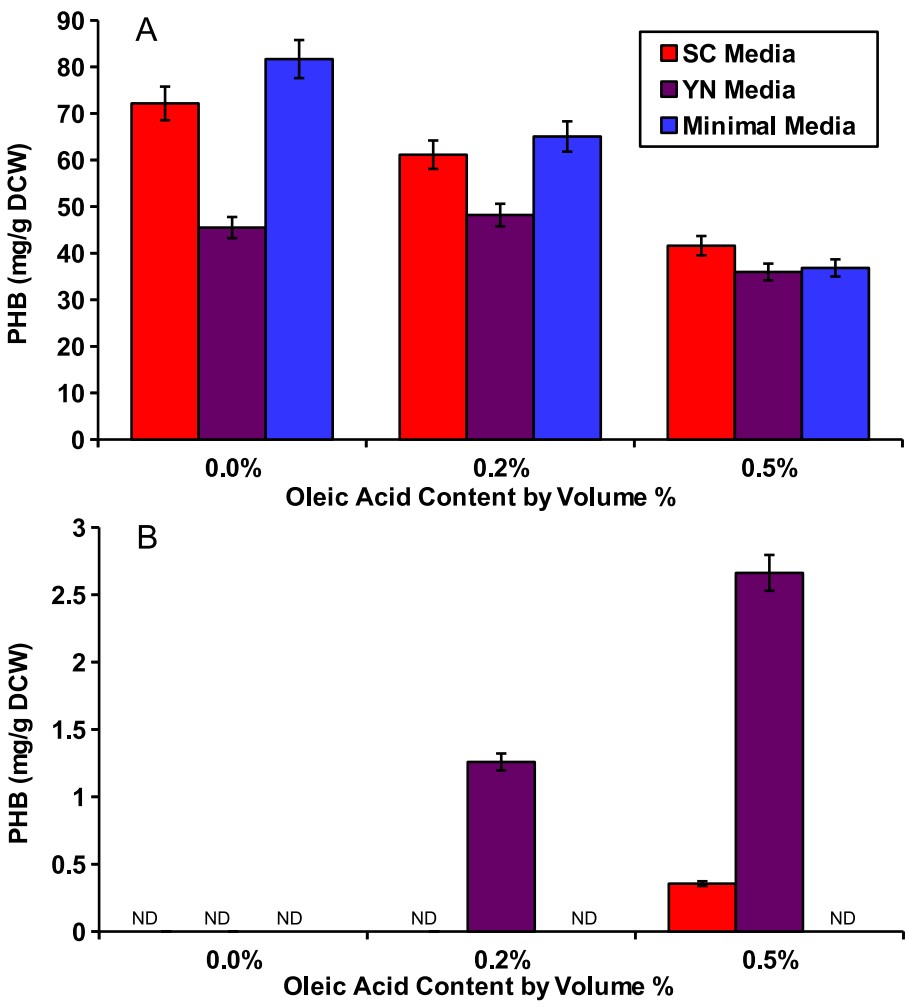

**Figure 2** **Oleic acid dependency of peroxisomal PHB.** Yeast strains with PHB producing enzymes were incubated for 48 hours in several media and with varied concentrations of oleic acid. (A) Yeast strain phbC containing the three pha enzymes without enzyme localization. (B) Yeast strain phbP with the three pha enzymes with localization tags. ND, Not Detectable. Error bars show estimated instrument error.

*aat2* and *ino1* are transcription factors that promote fatty acid synthesis or fatty acid utilization. Three of the genes, *rpd3*, *opi1* and *in3* are transcription factors that repress fatty acid synthesis. The genes *pot1* and *pox1* were selected as integral enzymes required for beta-oxidation. Genes *pex5* and *pex7* are peroxisome transport proteins responsible for import of peroxisome-specific proteins. The gene *inp2* was selected for its role in peroxisomal inheritance and the two genes *slt2* and *atg36* were selected for their role in pexophagy. The peroxisomal PHB enzymes were expressed in all thirteen knockout mutants and the wild-type yeast for five days in YN media at varied concentrations of oleic acid (Fig. 4). Beta oxidation gene knockouts Δ*pot1*, and Δ*pox1*, as well as peroxisomal transport protein knockouts Δ*pex5* and Δ*pex7* exhibited dramatic decreases in PHB production. Knockout strain Δ*aat2* showed relatively little change relative to the wild type

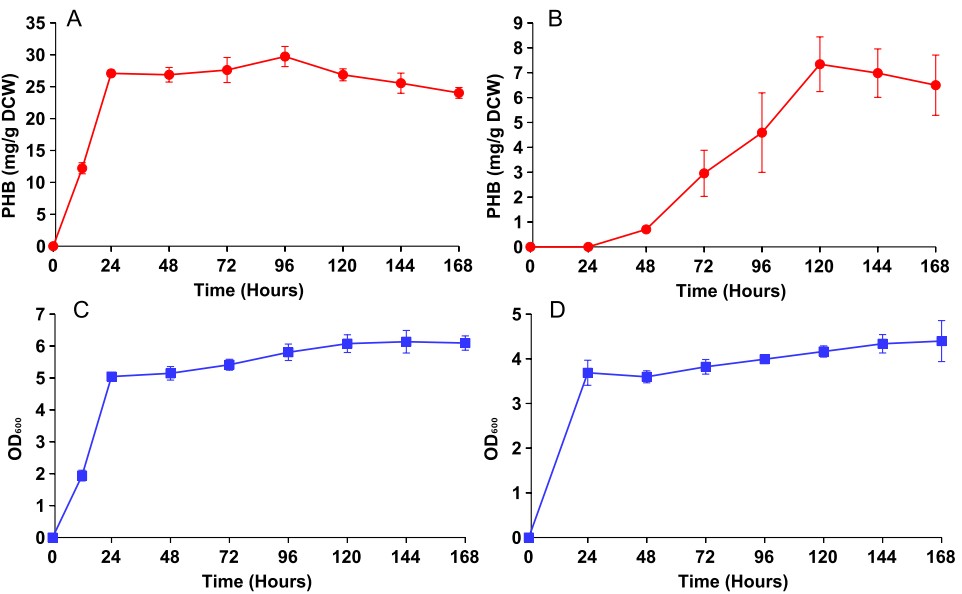

**Figure 3** **PHB production and growth curves for cytosolic and peroxisomal strains.** Both strains were cultured for five days in YN media with 0.5% oleic acid. (A) PHB content vs. time for the cytosolic strain phbC. (B) PHB content vs. time for phbP. (C) Growth curve for the phbC. (D) Growth curve for phbP. Error bars show standard deviation among triplicate cultures.

while Δ*adr1* exhibited decreased PHB production (Fig. 4A). None of the three fatty acid synthesis repressor knockouts (Δ*rpd3*, Δ*sin3* and Δ*opi1*) showed a strong increase in PHB production but Δ*sin3* and Δ*opi1* exhibited decreased production of PHB at 0.5% oleic acid relative to the wild type.

## DISCUSSION

Evidence for peroxisomal localization of the PHB producing enzymes is first demonstrated in Fig. 2. The cytosolic variant of the PHB pathway was not significantly affected by external oleic acid concentrations because acetyl-CoA in the cytosol was mostly derived from glucose. The peroxisomal variant of the PHB pathway, however, demonstrated a strong correlation between oleic acid content and PHB production, because acetyl-CoA in the peroxisome is mostly derived from beta-oxidation of fatty acids such as oleic acid (*Chen, Siewers & Nielsen, 2012*). There was also no detectable PHB produced from the peroxisomal pathway in Fig. 2 when oleic acid was absent, suggesting minimal leakage of cytosolic acetyl-CoA into the peroxisome. Therefore, it is appropriate to use PHB production from the compartmentalized biosensor to indicate relative peroxisomal acetyl-CoA availability.

Even with similar levels of glucose and oleic acid, the peroxisomal variant strain, phbP did not produce as much PHB as the cytosolic variant, phbC, within 48 hours (Fig. 2), so it was hypothesized that two days of incubation was not long enough for the yeast cells to fully switch from glucose utilization to fatty acid utilization (*Gurvitz & Rottensteiner, 2006*). Growth and production curves, shown in Fig. 3 demonstrate that PHB production in the phbP strain started only after 24 hours, which is the same amount of time it took to

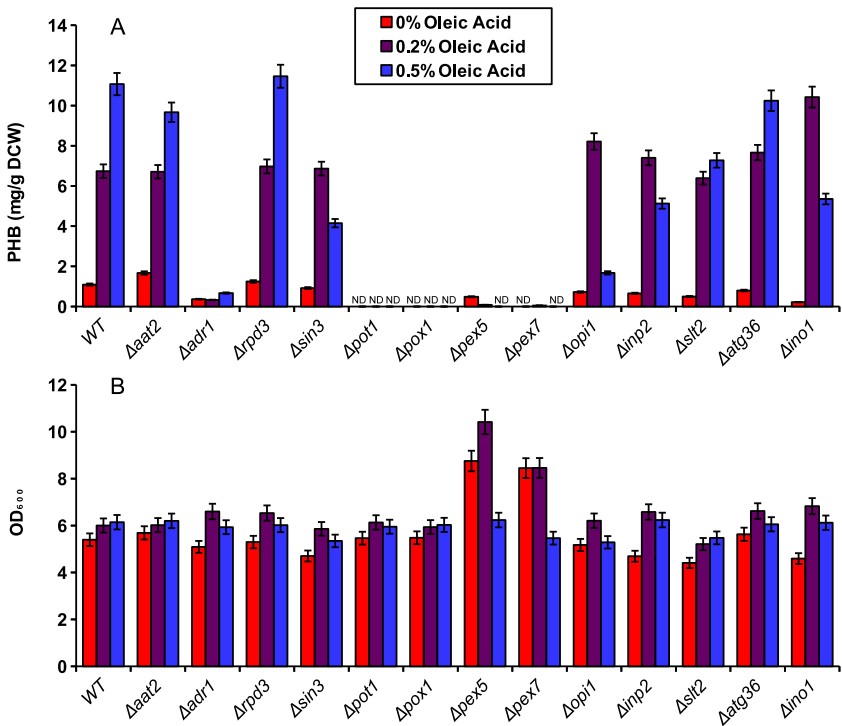

**Figure 4** **Screening of mutants using the peroxisomal acetyl-CoA assay.** Yeast knockout strains and the wild type (WT) were incubated for five days in YN media with various oleic acid concentrations. (A) PHB concentration. (B) Cell density (in $OD_{600}$) of the mutant screening cultures. ND, Not Detectable. Error bars show estimated instrument error.

reach stationary phase. This suggests that the yeast cells used glucose for cellular growth until it was depleted, then switched to beta-oxidation of oleic acid. However, the strain phbP was never able to generate as much PHB as the cytosolic control, phbC, even when given similar quantities of oleic acid and glucose and when given ample time to utilize oleic acid.

The compartmentalized biosensor was then applied to screening several single-gene knockouts for altered acetyl-CoA availability in the peroxisome. As expected, Δpot1, Δpox1, Δpex5 and Δpex7 all exhibited dramatic decreases in peroxisomal PHB production as those genes are necessary for beta oxidation. Similarly, Δadr1 reduced PHB production because of its role in activating fatty acid utilization. It is puzzling, however, that the Δpex5 strain did not exhibit high levels of cytosolic PHB production. Since pex5p (the transport protein coded for by pex5) is needed for peroxisomal localization of PTS1-tagged proteins, it was expected that the biosensor enzymes would accumulate in the cytosol as has been observed for PTS1 tagged yellow fluorescent protein (DeLoache, Russ & Dueber, 2016). However, the pex5 knockouts in this study did not perform similarly to the strains using enzymes without PTS1 tags. The reason for this is still unclear.

Other knockouts were predicted to increase peroxisomal acetyl-CoA. Fatty acids generated by the cell could be used to supplement the exogenous source of oleic acid

for beta-oxidation. As repressors of fatty acid synthesis, Δ*rpd3*, Δ*sin3* and Δ*opi1* had potential to increase PHB levels by providing additional fatty acids from the cytosol to be digested in the peroxisome. However, no dramatic increase in PHB was observed for these knockouts. One possible explanation is that PHB production was being limited by NADPH availability rather than peroxisomal acetyl-CoA. Further testing is needed to ensure sufficient NADPH availability and to determine the biosensor's full dynamic range.

Unlike the two-day cultures used for Fig. 2, the five-day cultures in the knockout experiments showed low levels of PHB production even in the absence of oleic acid. This could be caused by acetate being produced and converted to acetyl-CoA in the peroxisome by the pyruvate dehydrogenase bypass after the glucose has been depleted (*Nielsen, 2014*). PHB might also be produced by low, native availability of acetoacetyl-CoA and 3-hydroxybutyric-CoA intermediates. In a study by Leaf et al., yeast cells expressing only the final enzyme, PHB synthase, were able to produce PHB, albeit at only 1 mg/g DCW (*Leaf et al., 1996*). Intermediates produced in this manner would still be positively correlated to acetyl-CoA availability in the peroxisome, and thus, do not interfere with the intended function of the biosensor.

## CONCLUSIONS

In summary, we developed an assay that used three compartmentalized enzymes to convert peroxisomal acetyl-CoA to a reporter compound, PHB. By quantifying PHB levels, we successfully applied this compartmentalized biosensor to screen a library of yeast mutants and identified knockouts with drastically impaired PHB synthesis, which suggests the possibility that overexpression of some of those genes might result in increased acetyl-CoA availability in the peroxisome. Further testing is needed to ensure that PHB levels correlate quantifiably to acetyl-CoA availability, that NADPH availability does not become limiting and to ensure proper enzyme localization. Further testing is also needed to ensure that differences in PHB production are not caused by differences in enzyme expression among the different strains and culturing conditions used. Metabolic engineering of compartmentalized pathways requires measurement tools for pathway design and evaluation. Traditional methods for measuring metabolite concentrations are generally limited to cytosolic or whole-cell sources of the metabolite. In contrast, as shown in this study, compartmentalized biosensors could be a valuable addition to current synthetic toolkits on analyzing intracellular pools of metabolites.

This study illustrates only one specific application of a compartmentalized biosensor, but the strategy could be applied to other metabolites in other compartments. For example, the concentrations of acetyl-CoA in the mitochondria have been estimated to be 20-30 times higher than in the cytosol (*Weinert et al., 2017*) and a recent study involved re-localizing heterologous enzymes for the production of valencene to the mitochondria of *S. cerevisiae* via targeting signal peptides resulting in increased production compared to cytosolic expression of those same enzymes (*Farhi et al., 2011*). Further optimization of this localized pathway could benefit from a biosensor similar to ours to detect the critical precursor, farnesyl diphosphate (FDP). FDP can be detected using a fluorescent

enzymatic assay (*Dozier & Distefano, 2012*) and compartmentalization of this assay could be accomplished by introducing N-terminal mitochondrial localization signal (MLS) tags such as from subunit IV of the yeast cytochrome C oxidase (CoxIV) to the enzymes of the fluorescent assay (*Avalos, Fink & Stephanopoulos, 2013*).

We anticipate that compartmentalized biosensors can be used to better understand the compartmentalized metabolism in various organelles and facilitate the "design-build-test" cycle of compartmentalized metabolic engineering.

### Funding
This study was supported by a start-up fund (#175323) from Virginia Tech and the Junior Faculty Award of Institute for Critical Technology and Applied Science (#J0663185) from Virginia Tech. The funders had no role in study design, data collection and analysis, decision to publish, or preparation of the manuscript.

### Grant Disclosures
The following grant information was disclosed by the authors:
Virginia Tech and Junior Faculty Award of Institute for Critical Technology and Applied Science: #175323.
Virginia Tech: #J0663185.

### Competing Interests
The authors declare there are no competing interests.

### Author Contributions
- Herbert M. Huttanus conceived and designed the experiments, performed the experiments, analyzed the data, prepared figures and/or tables, authored or reviewed drafts of the paper, and approved the final draft.
- Ryan S. Senger analyzed the data, authored or reviewed drafts of the paper, and approved the final draft.

### Data Availability
The PHB concentration values are available as a Supplemental File.

### Supplemental Information
Supplemental information for this article can be found online at http://dx.doi.org/10.7717/peerj.9805#supplemental-information.

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
