# Peer review of "A synthetic biosensor to detect peroxisomal acetyl-CoA concentration for compartmentalized metabolic engineering"

_PeerJ, doi:10.7717/peerj.9805_

## Round 0.1 · original submission · Major Revisions

Thank you for submitting this interesting study. The two expert reviewers and I agree that the problem that you address is significant, and the approach you describe is potentially effective in addressing the aims of the study.

However, the reviewers both bring up an important deficiency in the work: there is no reported calibration of the response of the sensor to AcCoA abundance, either in the cytoplasm or the peroxisome. For example, Reviewer 1 states "...but some sort of validating measurements are needed, such as direct measurement of AcCoA and correlation with PHB titers." Reviewer 2 says "How can this be a sensor without calibration of acetyl-CoA levels? Normalization of expression in both the cytosol and peroxisome is necessary to support this claim."

Reviewer 2 also asks for more direct evidence that the sensor is specific to the peroxisome.

To be considered for acceptance, a revised manuscript must satisfactorily address these two points.

The reviewers also mention several smaller points that will enhance the clarity of the manuscript. Please address these points as you see fit.

I also noted in my reading that I was initially confused by the significance of having oleic acid present in the experiments described in Fig. 2. It was only when I reached the Discussion that this became clear to me. I suggest that you add the rationale for the use of oleic acid to the Introduction or Results sections (perhaps the sentences in the middle of the first paragraph of the Discussion), and perhaps add oleic acid to Fig. 1, as Reviewer 1 suggests, to assist the reader in understanding the experimental design.

Thank you again for submitting your work to PeerJ. I look forward to seeing a revised manuscript.

Reviewer 1 ·

Basic reporting

- The authors need to be consistent with convention in the references to genes (lowercase italics) and proteins (uppercase, not italics). For example, the abstract and the Figure 1 legend both refer to “three enzymes”, but the names are presented in non-italicized lowercase.
- It would be useful to have a table describing all strains in the main text.
- “Saccharomyces” can be abbreviated as “S.” after the first usage.
- It is strange (but not incorrect) that the x-axis in Figure 2 ranges from high oleate concentration to low concentration instead of the more conventional low  high presentation.
- Figure 3 – why not present both PHB data series in a single graph and both OD data series in a single graph? This would enable easier comparison between the two strains.

Experimental design

- On first glance, it is easy to misinterpret the experimental design. This reviewer first thought that PHB production was being used to demonstrate the effectiveness of the AcCoA-sensing biosensor, when actually PHB is the output of the biosensor. The authors should consider modifying Figure 1 to make this clearer. Also, oleate (since it is used throughout the manuscript) should be explicitly labeled in Figure 1.
- The only data presented as functionality of this biosensor is PHB production. The trends in the PHB titers are consistent with the expected trends of AcCoA abundance and of targeting to the peroxisome, but some sort of validating measurements are needed, such as direct measurement of AcCoA and correlation with PHB titers.
- Figure 2 – the authors state that the error bars refer to “estimated instrument error”. Does this mean that biological replicates were not assessed?

Validity of the findings

- The legend of Figure 1 refers to “rapid” recognition of the tags. How did the authors establish that the recognition is rapid?
- Line 187, the authors refer to “a significant effect”, but no supporting statistical analysis is described.

Additional comments

This manuscript describes the characterization of PHB production as a biosensor of AcCoA availability within the peroxisome. There is a strong demand for an ability to estimate AcCoA abundance within the peroxisome, and more generally, to detect specific metabolites within specific subcellular compartments. The authors detect strong, but indirect evidence of the functionality of their biosensor system. However, there manuscript is lacking a direct correlation of AcCoA abundance in the peroxisome and performance of this biosensor.

Reviewer 2 ·

Basic reporting

The language is mostly clear and unambigious. The literature is adequately referenced.

Comments:
Line 47 – What does efficiency refer to?
The word sensor is being misused through. The output is not easy to measure.
Figure 2 has data that should be listed as N.D. that is not.
Figure 3 – graphs should not have negative values on either axis.

Experimental design

The research question is well defined, relevant and meaningful. The experiments are incompletely designed, as it remains unclear to what extent the reporter is specific to the peroxisome. The methods are well described for replication.

Validity of the findings

The claim "With high levels of enzyme expression, we hypothesize that the production of the reporter compound will be rate-limited by the substrate concentration, allowing for relative quantification of the substrate." is not validated. In fact, the results showing that expected knockouts do not improve on the WT could be explained by te

More definitive proof is needed to show that enzymes and PHB are localized.

The claim "the PHB signal is representative of peroxisomal acetyl-CoA concentration" How can this be a sensor without calibration of acetyl-CoA levels? Normalization of expression in both the cytosol and peroxisome is necessary to support this claim. It also requires showing that the availability of acetyl-CoA is limiting, rather than NADPH.

How do you know that in low oleic acid, when peroxisomes are not abundant that Pha enzymes were efficiently translocated and didn’t remain in the cytosol?

Additional comments

N/A

---

## Round 0.2 · accepted · Accept

Your careful attention to the comments of the reviewers is appreciated. We look forward to the publication of this interesting study.

Reviewer 1 ·

Basic reporting

The authors have addressed the concerns raised in the previous round of review

Experimental design

The authors have addressed the concerns raised in the previous round of review

Validity of the findings

The authors have addressed the concerns raised in the previous round of review

Additional comments

The authors have addressed the concerns raised in the previous round of review